# Stereotactic Body Radiation Therapy in Patients with Oligometastatic Disease: Clinical State of the Art and Perspectives

**DOI:** 10.3390/cancers14051152

**Published:** 2022-02-23

**Authors:** Rémy Kinj, Emilien Muggeo, Luis Schiappacasse, Jean Bourhis, Fernanda G. Herrera

**Affiliations:** 1Service of Radiation Oncology, Department of Oncology, Lausanne University Hospital, 1010 Lausanne, Switzerland; emilien.muggeo@chuv.ch (E.M.); luis.schiappacasse@chuv.ch (L.S.); jean.bourhis@chuv.ch (J.B.); 2Service of Immuno-Oncology, Department of Oncology, Lausanne University Hospital and University of Lausanne, 1010 Lausanne, Switzerland

**Keywords:** oligometastasis, oligometastatic disease, metastasis directed therapy, radiotherapy, SBRT

## Abstract

**Simple Summary:**

Stereotactic radiation therapy (SBRT) is a type of radiation therapy in which a small number of high doses of radiation are delivered to a target volume using highly accurate equipment in order to maximize cancer control while minimizing side effects on healthy tissues. SBRT’s precise role varies according to the primary location and subtype of the oligometastatic state. The purpose of this review is to clarify the role of SBRT in various cancer types and to define its position based on the oligometastatic disease state.

**Abstract:**

Stereotactic body radiation therapy (SBRT) is a form of radiation therapy (RT) in which a small number of high doses of radiation are delivered to a target volume using highly sophisticated equipment. Stereotactic body radiation therapy is crucial in two cancer stages: early primary cancer and oligometastatic disease, with the goal of inducing complete cancer remission in both. This treatment method is commonly used to treat a variety of disease types. Over the years, a growing body of clinical evidence on the use of SBRT for the treatment of primary and metastatic tumors has accumulated, with efficacy and safety demonstrated in randomized clinical trials. This article will review the technical and clinical aspects of SBRT according to disease type and clinical indication.

## 1. Introduction

Hellman and Weichselbaum first proposed the concept of oligometastatic disease in 1995 [1]. An oligometastatic disease is a stage of the disease that is intermediate between locoregionally advanced and metastatic disease and is still treatable curatively.

De novo oligometastasis, oligo-recurrence, oligo-progression and oligo-persistence are the four categories of oligometastatic disease, corresponding to the different clinical scenarios that capture the spectrum of oligometastatic disease [2,3]. The term “de novo” oligometastasis refers to newly diagnosed cancer with few metastases occurring concurrently with the primary tumor. Oligo-recurrence refers to patients who have been treated for metastatic disease and have a relapse in a few new metastatic sites. Oligo-progression refers to patients who are controlled by systemic treatments and progress only on a few metastatic sites, whereas oligo-persistence refers to patients who respond to systemic treatment but still have a few metastatic sites.

It is important to note that a patient can experience dynamic transitions between oligo-recurrent, oligo-progressive, and oligo-persistent disease based on response to local and systemic therapy. For example, in a group of patients with oligometastatic prostate cancer, a median of four courses of radical local treatment were required over the course of the metastatic disease. As a result, the transition from one oligometastatic state to another is not always indicative of disease progression, but rather of a really limited oligometastatic phenotype.

Ablative treatments for oligometastases must be as curative as possible and may include local surgery, radio-ablations and stereotactic body radiotherapy (SBRT).

It is critical to note that SBRT has gradually been proposed as an alternative to metastasectomy and other ablative treatments. SBRT is an image-guided RT technique that delivers high doses with high precision to small target volumes in a single or small number of fractions while minimizing radiation exposure to non-targeted tissue.

Before treatment can be administered, the location of the tumor or target volume must be confirmed, and immobilization devices must be used to keep the patient precisely in the same position throughout the treatment delivery [4,5]. Extracranial SBRT necessitates the use of appropriate RT devices that allow for a tight gradient of dose and a rapid decrease in dose to maximize healthy tissue organ sparing in order to guarantee maximum normal tissue spearing [6,7] (Figure 1).

From a biological point of view, in addition to direct cytotoxicity, SBRT may introduce a new mechanism of radiation-induced damage, involving microvascular damage and endothelial apoptosis, resulting in microvascular disruption and death of the tissue irrigated by that vasculature [8]. Stereotactic body radiation therapy, in addition to its vasculature remodeling effect, can induce a potent “in situ” vaccination effect capable of inducing T cell infiltration as a result of high antigen load [9,10].

Extracranial SBRT can be delivered to various involved organs, such as the lung, bone, liver, or adrenal glands using a variety of SBRT regimens and techniques.

The multicentric “Stereotactic ablative radiotherapy versus standard of care palliative treatment in patients with oligometastatic cancers: (SABR-COMET)” phase II randomized trial now serves as a proof-of-concept for the benefit of metastasis guided SBRT in oligometastatic patients. In this trial, patients presented oligometastasis from a variety of primaries, including the colon, lung, breast, and prostate, and the sites of irradiated lesions included the lung, bone, liver, adrenal and others. Patients in the SBRT group received stereotactic radiation to all sites of metastatic disease, with the goal of achieving disease control while minimizing potential toxicities. The standard arm was the best supportive care. This trial reported a survival benefit of metastases-directed SBRT for oligometastatic patients (1–5 metastatic sites) who had their primary malignancy under control (median OS 28 months in the control group, 95% CI: 19–33 vs. 41 months in the SBRT group, 95% CI: 26-not reached; HR 0.57, 95% CI: 0.3–1.1; *p* = 0.09) [11]. Importantly, the overall survival (OS) benefit became larger in magnitude after a longer follow-up of 5 years, with 18% (95% CI, 6–34%) in control group vs. 42% (95% CI: 28–56%) in SBRT group (*p* = 0.006) [12].

In terms of safety, Lehrer et al. reported a meta-analysis on 943 patients (1290 lesions) in which the estimate for late grade 3 or more adverse events was 1.2% after a median follow-up of 16.9 months. These are acceptable levels of toxicity, and validation is being performed in prospective clinical trials [13].

Clinical experience and challenges in a variety of disease types are reviewed and discussed in this paper. Because the primary location is one of the most important factors influencing patient outcomes, we will discuss the findings of prospective trials that focused on specific primary sites, evaluating the outcomes of SBRT for patients with oligometastatic disease by disease type.

We concentrated on the new classification of de novo, oligometastatic, oligorecurrence, and oligoprogressive disease in order to better define the strategies in the various disease settings [2,3]. Figure 2 depicts the most common types of oligometastatic disease treated with SBRT and usual doses and fractionations (Tables 1–4, Figure 2).

## 2. Materials and Methods

We conducted a systematic literature review (registration number: reviewregistry1306) based on PubMed and adhered to PRISMA guidelines [14]. Two investigators (RK and FH) searched the databases independently and until 11 November 2021. The search terms were: ([oligometastasis OR oligometastases OR oligorecurrence OR oligoprogressive OR oligopersistent] AND [radiotherapy]). The results were then filtered using the following criteria: “<10 years”. We looked at prospective and retrospective trials and therapeutic interventional studies that report outcomes, such as OS, progression-free survival (PFS), or disease recurrence. Articles were excluded if they did not correspond to our review topic, if they only reported quality-of-life or if they were only about brain metastases. We prioritized prospective trials and meta-analyses to be described in the main text. We identified 972 articles that match our search terms. After applying filters, we identified 243 papers. After prioritizing, we selected 66 articles to be mentioned in the clinical results part.

## 3. Prostate Cancer

Prostate cancer (PCa) is the world’s second most common cancer in men and the sixth leading cause of cancer-related death. Androgen deprivation therapy (ADT) is frequently the treatment of choice for patients who have been diagnosed with metastatic or locally advanced PCa for the first time. ADT is usually combined with abiraterone, docetaxel, apalutamide, or enzalutamide in men with castration sensitive metastatic PCa. The agent chosen will be determined by the risk and clinical burden of the disease, as well as the patient’s comorbidities. Metastatic castration sensitive disease has also been divided by tumor burden; a high burden of disease has included the presence of visceral metastases, a bone-metastasis burden classified by site (beyond the axial skeleton), or a high number of lesions (more than five), or a combination of these [15,16].

Patients with metastatic castration-resistant prostate cancer (mCRPC) frequently develop distant metastasis, and bony metastases can result in significant morbidity and a decline in quality of life. Individual patient data from 8820 men with mCRPC who were treated with a docetaxel-containing regimen as part of one of nine phase III trials [17] showed that OS was highest in those with lymph node-only disease and gradually declined in those with bone, lung, or liver metastases (median 31.6, 21.3, 19.4, and 13.5 months, respectively). As a result, men who have rising prostate-specific antigen (PSA) after ADT, but no evidence of macroscopic metastatic disease are classified as having non-metastatic CRPC. Multiple agents, all given in conjunction with continued ADT, have been shown in phase III trials to improve OS in men with mCRPC, and include abiraterone, enzalutamide, apalutamide, darolutamide; chemotherapy: docetaxel, cabazitaxel and immunotherapy: sipuleucel-T (in minimally symptomatic men who have a slowly progressive disease).

Based on the data presented above, it is critical for PCa patients to identify intermediate stages of disease dissemination that can benefit from either improved systemic therapies or metastasis-directed therapies (MDT). Oligometastatic PCa is a broad term that encompasses at least three distinct entities, each with its own set of biological signatures and behavior. i) De novo oligometastasis refers to a distinct group of patients with PCa who have spread to limited areas prior to any definitive therapy; ii) Oligo-recurrent PCa refers to the development of limited sites of distant dissemination following primary radical prostatectomy (RP) or radiotherapy and iii) Oligo-progressive PCa refers to patients who gradually progress on less than three to five lesions despite continued systemic therapy [18] (Table 1, Figure 2).

### 3.1. De Novo Oligometastatic Prostate Cancer

Local ablative therapy to the prostate is the most commonly used treatment in the de novo oligometastatic setting, and it has been studied in two randomized trials, which serve to reinforce the idea that treating the oligometastatic stage with ablative therapies is beneficial.

The phase III HORRAD trial randomly assigned 432 men with primary metastatic PCa with bone metastases, as well as a serum PSA > 20 ng/mL, to ADT with or without external beam RT (70 Gy in 35 daily 2 Gy fractions). Two-thirds of the men had more than five bone metastases. The addition of radiation did not improve OS (the primary endpoint), but it did prolong the median time to PSA progression (median 15 vs. 12 months, HR = 0.78, 95% CI: 0.63–0.97, *p* = 0.02). Men with fewer than five metastases had a better chance of survival, according to an unplanned subgroup analysis, but the result was not statistically significant (HR = 0.68, 95% CI: 0.42–1.10) [19]. Similarly, 2061 men with newly diagnosed metastatic PCa were randomly assigned to ADT with or without docetaxel and with or without prostate radiation in the phase III STAMPEDE trial, (which could be either 36 Gy in six consecutive weekly fractions of 6 Gy or 55 Gy in 20 daily fractions of 2.75 Gy over four weeks). Metastatic burden was assessed at randomization using whole-body scintigraphy and computed tomography or magnetic resonance imaging staging scans, and it was classified using the CHAARTED trial definitions [20,21]. Overall survival (the primary endpoint) was not improved by prostate irradiation, but three-year failure-free survival was 32 vs. 23%, (HR = 0.76, 95% CI: 0.68–0.84, *p* < 0.0001). Prostate RT improved OS in men with a low metastatic burden (three-year survival 81 vs. 73%, HR = 0.68, 95% CI: 0.52–0.90) but not in those with a high metastatic burden (HR = 1.07, 95% CI: 0.90–1.28). The acute adverse effects of prostate irradiation were minor, with only 5% reporting grade 3 or 4 bladder or bowel events, compared with 1% in the control group. Approximately 1% of men who had prostate irradiation experienced late grade 3 or higher gastrointestinal toxicity, whereas none did in the control group.

The pooled results of both trials, on the other hand, concluded that there was an overall improvement in biochemical PFS (HR = 0.74, 95% CI: 0.67–0.82, *p* < 0.00001) and failure-free-survival (HR = 0.76, 95% CI: 0.69–0.84, *p* < 0.00001), which translated into an approximately 10% benefit at three years for the entire cohort. In unplanned subgroup analysis of the STAMPEDE randomized trial, an OS benefit was observed in the group with three or fewer bone metastases (three-year OS 75 vs. 85%, HR = 0.64, 95% CI: 0.46–0.89) but not in those with four or more bone metastases (three-year OS 53 vs. 52%, HR = 1.12, 95% CI: 0.93–1.34). Prostate irradiation was of no benefit in patients with visceral or other metastases [22].

Concerning the combination of RT with immunotherapy, Fizazi et al. explored the impact of RT (a single dose of 8 Gy,) on bone metastases (one to five metastases) followed by ipilimumab or placebo in men with mCRPC (who received docetaxel previously). The primary endpoint was OS and 799 patients were randomized. In long-term analysis the RT and immunotherapy arm presented an OS benefit of 7.4%, 6.8%, and 5.2% at 3, 4 and 5 years, respectively (HR = 0.66, 95% CI: 0.52–0.84) [23].

A North American phase 2 trial is designed to test a comprehensive systemic and tumor directed therapeutic strategy for patients with newly diagnosed de novo oligometastatic PCa. Patients with newly-diagnosed M1a/b PCa and 1–5 radiographically visible metastases (excluding pelvic lymph nodes) are being treated locally with RP, six months systemic therapy (leuprolide, abiraterone acetate with prednisone, and apalutamide), metastasis-directed SBRT, and post-operative fractionated RT to the primary tumor bed if pT ≥ 3a, N1, or positive margins are present. The primary endpoint is the percentage of patients who achieve a serum PSA of <0.05 ng/mL six months after recovery of serum testosterone ≥150 ng/dL (NCT03298087, ClinicalTrials.gov) [24].

The results of these trials will help us to design the next generation of clinical trials based on the concept of maximal cytoreduction.

### 3.2. Oligo-Recurrent PCa

Ost et al. provided the first prospective evidence in a phase II study in which patients with oligorecurrent PCa and three extracranial metastases on choline positron emission tomography computed tomography (PET/CT) were randomly assigned to either PSA surveillance every 3 months (*n* = 31) or metastasis-directed therapy (MDT, surgery or SBRT) to all lesions (*n* = 31), with the goal of improving ADT–free survival. ADT was initiated for symptomatic or local progression, or when more than three metastases developed. After 3 years of median follow-up, the interventional group’s ADT-free survival was 21 months compared to 13 months in the control group (HR = 0.60, 80% CI: (0.40–0.90); *p* = 0.11). Toxicity was low, with only six patients in the MDT arm suffering from grade 1 toxicity. There was no evidence of toxicity grade 3 or higher. The authors concluded that for oligorecurrent PCa, ADT-free survival was longer with MDT than with surveillance alone, implying that MDT should be investigated further in phase III trials. Although these findings highlight the potential of MDT to delay the initiation of systemic therapy and its associated side effects, there was no statistically significant improvement in 1-year quality-of-life, possibly due to a lack of power to detect such a difference [25].

An interim analysis of the phase II TRANSFORM non-randomized single institution trial looked at men who had relapsed with up to five lesions after definitive local treatment for primary PCa. The goal was to determine the proportion of patients who did not require systemic treatment after metastasis-directed SBRT. In total, 199 patients were enrolled in the study to receive fractionated SBRT (10 fractions of 5 Gy each) to all visible lesions. The authors defined the primary endpoint as the start of ADT for hormone naïve patients and the start of second-line ADT or chemotherapy for those who had prior ADT; 51.7% of patients did not require systemic therapy 2 years after SBRT (95% CI: 44.1–59.3). Over the entire follow-up period, the median length of treatment-free survival was 27.1 months (95% CI: 21.8–29.4). There was no difference in the efficacy of SBRT when treating 4–5 vs. 1–3 lesions. In 75% of patients, PSA was reduced with PSA levels felling to an undetectable level in six patients. There were no grade 3 or higher toxicities observed. The authors concluded that these interim results suggest that SBRT can be used to treat up to five synchronous PCa oligometastases to delay systemic therapies [26].

Siva et al. published the findings of a single arm prospective clinical trial that investigated the safety and feasibility of single fraction SBRT for patients with oligometastatic PCa. Thirty-three consecutive patients were followed for 2 years after receiving a single dose of 20 Gy SBRT to a total of 50 lesions. Twenty patients had only bone disease, 12 had only node disease, and one had both. There was one grade 3 adverse event that was a vertebral fracture that required spinal instrumentation. The one- and two-year local control (LC) was 97% (95% CI: 91–100) and 93% (95% CI: 84–100), PFS was 58% (95% CI: 43–77) and 39% (95% CI: 25–60), respectively. The two-year freedom from ADT was 48%. The authors concluded that the SBRT approach was safe and that half of the patients in the study avoided ADT at 2 years [27].

Several ongoing trials are looking into the possibility of combining local treatment with metastasis-directed RT in patients presenting oligo-recurrent PCa. This question is hypothesized in the PEACE-V trial. Patients diagnosed with Prostate-specific membrane antigen (PSMA) PET/CT detected pelvic nodal oligorecurrence (≤5 nodes) following radical local treatment will be randomized in a 1:1 ratio to arm A: MDT and 6 months of ADT, or arm B: whole pelvis RT added to MDT and 6 months of ADT. The primary endpoint is metastasis-free-survival, the estimated study completion is 31 December 2023 (NCT03569241, ClinicalTrials.gov) [28].

The ORIOLE trial is a phase II randomized study evaluating the safety and efficacy of SBRT in oligometastatic hormone-sensitive PCa. Fifty-four men with oligometastatic prostate adenocarcinoma will be randomized 2:1, the primary endpoint will be PFS, the study completion date is expected mid-2023 (NCT02680587, ClinicalTrials.gov) [29].

### 3.3. Oligo-Progressive PCa

To date, no prospective trial results in patients with oligo-progressive or oligo-persistent PCa have been published.

Triggiani et al. conducted a retrospective study to differentiate the results of SBRT in patients with oligorecurrent PCa from those with oligoprogressive PCa. Over 100 patients were treated with SBRT for 70 lesions, 41 of whom had oligoprogressive PCa. Progression-free-survival seemed comparable between the two study populations, the median PFS was 17.7 months in oligo-recurrent PCa and 11 months in oligo-progressive PCa. Oligoprogressive patients experienced a 2-years LC of 90.2% with no grade ≥3 toxicity. The median distant PFS was 11 months and the median second-line systemic treatment-free survival was 22 months [30].

Another retrospective study looked at the outcomes of SBRT in a group of 68 patients with oligo-progressive mCRPC. Sixty-eight patients (112 lesions) were included in the study. The median time to PSA failure was 9.7 months, the time to the next intervention was 15.6 months, and the distant metastasis-free survival time was 10.8 months. When compared to a cohort of patients treated at the same institution and who only received a switch in systemic treatment but no SBRT (*n* = 52), SBRT was associated with a longer median time to PSA failure although the difference was not statistically significant (9.7 vs. 4.2 months, *p* = 0.066) [31].

**Table 1 cancers-14-01152-t001:** Main results of SBRT in oligometastatic PCa.

Author/Year	N^o^ of Patients/ Primary/Oligometastatic State	Phase/Design/N^o^ of Lesions	Arms (Investigational vs Control)	Primary Endpoint	Median PFS (Months)	Median OS (Months)	Toxicity (≥G3)
**Prostate Cancer**							
Boevé et al./2018 [19]	432 *De novo* OMD	Phase III Randomized ≤5 lesions	Prostate EBRT+ ADT vs ADT	OS	15 vs. 12	45 vs. 43	NA
Parker et al./2018 [20]	2061 *De novo* OMD	Phase III Randomized	Prostate EBRT+ ADT vs. ADT	OS	26 vs. 21	48 vs. 48	39% vs. 38%
Ost et al./2017 [25]	62 Oligo-recurrent	Phase II Randomized ≤3 lesions	MDT vs. Surveillance	ADT-free survival	21 vs. 13	NA	0% vs. 0%
Bowden et al./2020 [26]	199 Oligo-recurrent	Phase II ≤5 lesions	SBRT-MDT	Treatment escalation	27	NA	0%
Siva et al./2018 [27]	33 Oligo-recurrent	Phase II ≤3 lesions	SBRT-MDT	Feasibility	24	NA	3%
Triggiani et al./2017 [30]	141Oligo-recurrent/Oligo-progressive	Retrospective ≤3 lesions	SBRT	Distant FS	At 12 months In Oligo-recurrent 64.4% In Oligo-progressive 43.2%	NA	0%
Deek et al./2021 [31]	68Oligo-progressive	Retrospective ≤5 lesions	SBRT	PFS	10	NR	0%

ADT: androgenic deprivation treatment; N: number; OMD: oligometastatic disease; EBRT: external body radiotherapy; SBRT stereotactic radiotherapy; MDT: metastasis directed treatment; NA: not available; NR: not reached.

Prospective trials are enrolling patients to determine the role of SBRT in the treatment of oligo-progressive PCa. The OLI-CR-P is a prospective randomized phase II study that compares the safety and efficacy of metastasis-directed SBRT to observation in patients with oligo-progressive mCRPC (NCT04141709, ClinicalTrials.gov).

The TRAP trial is a multicenter, single-arm, phase II study that enrolls oligoprogressive androgen-suppressed PCa patients to evaluate the benefit of SBRT when combined with enzalutamide or abiraterone in terms of PFS (NCT0344303, ClinicalTrials.gov).

To the best of our knowledge, no trial is currently underway that attempts to study SBRT in a population of oligo-persistent PCa patients, leaving an open path for studying the role of SBRT in this setting.

## 4. Non Small Cell Lung Cancer

At the time of the diagnosis, more than half of non-small cell lung cancer (NSCLC) patients are metastatic. While metastatic lung cancer has traditionally been associated with poor survival, it has become clear in recent decades that metastatic lung cancer is a heterogeneous population with varying outcomes based on the extent and location of metastatic deposits. Furthermore, advances in imaging technology and the increased use of modalities, such as brain magnetic resonance imaging (MRI) and PET/CT have allowed for more accurate staging of lung cancer patients and the detection of previously undetected metastases. Patients with oligometastatic lung cancer have been found to have better survival outcomes than patients with more widely metastatic disease, and they account for up to 25–50% of all metastatic lung cancer cases [32]. Advances in metastatic NSCLC targeted systemic therapies, such as epithelial growth factor receptor (EGFR) inhibition and immunotherapy, have improved survival outcomes, emphasizing the importance of long-term LC of metastatic deposits. Patterns of failure analyses indicate that the most likely locations of failure following first-line chemotherapy are the initially involved sites, providing additional support for MDT [33].

The lung cancer group of the European Organization for research and Treatment of Cancer (EORTC) has agreed to define synchronous oligometastatic disease (sOMD) and to use it to classify patients in future clinical trials [34]. A maximum of five metastases and three affected organs were proposed in the definition, while the involvement of mediastinal lymph nodes was not considered. This definition necessitates the use of fluorodeoxyglucose (^18^F-FDG) PET/CT and brain imaging (preferably an MRI) to rule out the location of metastatic disease. Solitary liver metastasis should be investigated with MRI, while solitary pleural metastasis necessitates video-assisted thoracoscopy and biopsies of distant homolateral pleural locations. The minimum requirement for metastatic staging is ^18^F-FDG PET/CT, and histological confirmation is recommended if it affects the radiation treatment plan. When a radical disease-modifying therapy (which results in long-term disease control) is technically feasible for all tumor sites, has low toxicity, and can be offered to a patient, the term and definitions of sOMD must be used (Table 2, Figure 2 and Figure 3).

### 4.1. De Novo or sOMD NSCLC and Lung Directed Therapy

Two randomized trials studied the role of local consolidative RT in patients presenting oligometastatic NSCLC [35,36]. Gomez et al. conducted a multicenter randomized phase II trial in which patients with stage IV NSCLC and ≤3 metastatic sites after first line systemic therapy (platinum doublet) were randomly assigned to MDT (SBRT or surgery) in combination with maintenance systemic therapy or maintenance systemic treatment alone. Maintenance treatment consisted of four cycles of platinum doublet or three months of EGFR or ALK inhibitors (for patients with these specific mutations) [35]. After the randomization of 49 patients, the trial was terminated early at the interim analysis due to futility. There were 25 patients in the local consolidative therapy group and 24 in the maintenance treatment group. The median PFS for local consolidative therapy was 11.9 months (90% CI: 5.7–20.9) vs. 3.9 months (90% CI: 2.3–66) for maintenance treatment (HR = 0.35, 90% CI: 0.18–0.66, *p* = 0.0054). In any of the groups, there was only two grade 3 toxicity with no grade 4 adverse events or treatment-related deaths. This landmark trial proved that LCT plus/maintenance therapy improved PFS compared to maintenance therapy alone in patients with ≤3 NSCLC metastases that did not progress after initial systemic therapy.

Iyengar et al. conducted a single-institution randomized phase 2 study comparing maintenance chemotherapy alone to SBRT followed by maintenance chemotherapy for patients with limited metastatic NSCLC [36]. Patients had to have tumors that did not have EGFR- or ALK-targetable mutations but achieved a partial response or stable disease after induction chemotherapy. Patients were irradiated on the primary site as well as up to five metastatic sites. The primary endpoint was PFS. A total of 29 patients were enrolled in the study; 14 were allocated to the SBRT-plus-maintenance chemotherapy arm, and 15 to the maintenance chemotherapy–alone arm. The trial was terminated early after an interim analysis revealed a significant improvement in PFS in the SBRT-plus-maintenance chemotherapy arm of 9.7 months vs. 3.5 months in the maintenance chemotherapy–alone arm (*p* = 0.01). The toxicity was comparable in both arms, with two grade 3 toxicities in the maintenance arm alone and four grade 3 toxicities in the SBRT-plus-maintenance arm.

A meta-analysis of 21 studies looked into the addition of local thoracic RT to standard-of-care systemic treatment in patients with sOMD NSCLC [37]. The median OS and PFS were 20.4 and 12 months, respectively. The pooled 1-2-3 and 5-year OS rates were 70.3%, 43.5%, 29.3% and 20.2%, respectively. The addition of thoracic RT improved OS (HR = 0.44, 95% CI: 0.32–0.6; *p* < 0.001). Similarly, adding RT to the primary tumor improved PFS (HR = 0.42, 95% CI: 0.33–0.55; *p* < 0.001).

These trials shared the same belief in aggressive local treatment for patients with a low metastatic burden. However, additional research should be conducted to confirm this data ideally in phase III randomized studies. Gomez et al. and Iyengar et al. opened the path for randomized trials for evaluating the impact of localized treatment for oligometastatic NSCLC. Furthermore, because these trials were conducted prior to the immuno-oncology era, the same questions should be investigated with the inclusion of novel systemic agents, such as immunotherapy.

Bauml et al., conducted a single arm phase 2 study that enrolled 51 patients with sOMD and metachronous NSCLC (less than four metastatic lesions) after first line chemotherapy to evaluate the effect of SBRT or surgery following 4 weeks after treatment with an anti-PD-L1 immune checkpoint inhibitor (pembrolizumab, 200 mg every 21 days). Patients were not selected based on PD-L1 status but 34% had results positive for PD-L1 (≥1%) and 52% had CD8 T-cell infiltration of greater than 2.5%. After a median follow-up of 25 months, there was a statistically significant improvement in median PFS from historical control from 6.6 months to 19.1 months (95% CI: 9.4–28.7 months; *p* = 0.005). Progression was local only (at a site of a prior SBRT) in two patients, systemic only (outside SBRT volume) in 15 patients, and both in six patients [38].

This information led Theelen et al. [39] to perform the PEMBRO-RT trial, a randomized phase II study that included 92 patients with advanced stage NSCLC. The goal of this trial was to assess whether the addition of SBRT to a single tumor lesion prior to pembrolizumab enhances response in stage IV NSCLC patients. Ninety-two patients were randomly assigned to receive either pembrolizumab (200 mg/kg every 3 weeks) administered alone or after SBRT to a single tumor lesion until progression, unacceptable toxicity, or a maximum of 24 months. In the SBRT arm, the first pembrolizumab dose was given ≤7 days after completion of SBRT, consisting of three doses of 8 Gy delivered on alternate days to a single tumor site. The 3-month response rate was 18% in the control arm vs. 36% in the SBRT arm (*p* = 0.07). Median OS was 7.6 months in pembrolizumab alone vs. 15.9 months in the SBRT arm, (HR = 0.66; 95% CI: 0.37–1.18; *p* = 0.16). A significant improvement (64% vs. 40%; *p* = 0.04) was observed in the disease control rate at 12 weeks in the SBRT arm. 

The 14–18 CHESS-ETOP trial (NCT03965468) is a multicenter single arm phase II study designed to evaluate the efficacy of immunotherapy, chemotherapy and SBRT to metastases followed by definitive surgery or RT to the primary tumor, in patients with sOMD NSCLC. The primary endpoint is PFS. Patients will benefit from definitive primary treatment (surgery or curative radio-chemotherapy), SBRT to all oligometastatic sites and maintenance durvalumab for a maximum of 1 year until progression. The first patient was treated in November 2019, 47 patients are planned to be enrolled. If the results are positive, it could serve as yet another argument in favor of this approach in the context of combinatorial chemo-immunotherapy.

### 4.2. Oligo-Progressive NSCLC

A prospective phase II non-randomized trial involving 25 patients with oligoprogressive disease looked into the role of SBRT in patients who presented extra-cranial oligometastasis and EGFR mutated NSCLC receiving erlotinib. SBRT to progressive sites with tyrosine-kinase inhibitors (TKI) maintenance resulted in a 6 months median PFS and 29 months median OS (95% CI: 21.7–36.3) [40]. Similarly, in EGFR mutant NSCLC, when local SBRT was added to oligo-progressive lesions in a multi-institutional phase II trial, median PFS and OS were significantly greater of 15 months, and 20 months, respectively, than historical controls receiving systemic drugs alone. However, two patients had grade 3 toxicities related to SBRT (pneumonitis and back pain) [41].

Weickhardt et al. reported the retrospective results of 24 patients with oligo-progressive NSCLC who after initial progression on targeted therapy continued to receive the same drug in conjunction with SBRT. The disease control benefit was 6.2 months compared to the continuation of the drug alone. [42].

Chan et al. identified 25 patients who received SBRT for three or fewer oligo-progressive lesions and continued their systemic treatment with oral TKI. The results were compared to those of a group of patients with oligo-progressive NSCLC who received a systemic line switch. The study concluded that metastasis-directed SBRT provided an OS advantage of 10 months (28.2 vs. 14.7 months) compared to a switch of systemic therapy. Only one patient presented a grade ≥3 toxicity after RT [43]. 

Another retrospective study in a similar population of 46 patients with oligo-progressive NSCLC with druggable mutations found that after local SBRT and continuation of the TKI, the PFS was 7 months with no grade ≥4 [44].

**Table 2 cancers-14-01152-t002:** Main results of SBRT in oligometastatic NSCLC.

Author/Year	N^o^ of Patients/ Primary/Oligometastatic State	Phase/Design/N^o^ of Lesions	Arms (Investigational vs. Control)	Primary Endpoint	Median PFS (Months)	Median OS (Months)	Toxicity (≥G3)
Gomez et al./2016 [35]	74 De novo OMD	Phase II Randomized ≤3 lesions	Lung consolidative treatment vs. Maintenance	PFS	12 vs. 4	41 vs. 17	20% vs. 8%
Iyengar et al./2018 [36]	29 De novo OMD	Phase II Randomized ≤5 lesions	SBRT-MDT vs. Maintenance	PFS	10 vs. 4	NR	13% vs. 10%
Bauml et al./2019 [38]	51 De novo OMD	Phase II ≤4 lesions	SBRT-MDT + Pembrolizumab	PFS	19	NA	2%
Weiss et al./2019 [40]	25 Oligo-progressive	Phase II ≤3 lesions	SBRT-MDT	PFS	6	29	4%
Iyengar et al./2014 [41]	24 Oligo-progressive	Phase II ≤6 lesions	SBRT-MDT	PFS	15	20	8%
Weickhardt et al./2012 [42]	51 Oligo-progressive	Retrospective	SBRT-MDT	PFS	10	NA	NA
Chan et al./2017 [43]	50 Oligo-progressive	Retrospective Match cohort ≤ 3 lesions	SBRT-MDT vs. Chemotherapy	OS	7 vs. 4	28 vs. 15	4%
Qiu et al./2017 [44]	46 Oligo-progressive	Retrospective ≤ 5 lesions	MDT	PFS	7	13	22%

N: number; OMD: oligometastatic disease; SBRT stereotactic radiotherapy; MDT: metastasis directed treatment; NA: not available; NR: not reached.

Several prospective trials are currently enrolling patients with oligo-progressive NSCLC in order to refine the effect of SBRT and its impact on OS, PFS and quality of life.

The STOP trial is a phase II trial in which patients with oligo-progressive NSCLC are randomly assigned to receive standard of care systemic therapy plus SBRT to all sites of progressive disease or standard-of-care systemic therapy alone. A total of 90 participants are expected to be enrolled until June 2022 (NCT02756793, ClinicalTrials.gov).

HALT is a multicenter phase II/III trial that aims to enroll patients with mutation-positive advanced NSCLC who are receiving targeted therapy and have oligo-progressive disease. A total of 110 patients are expected to be enrolled and randomly assigned 2:1 to either SBRT or no SBRT (NCT03256981, ClinicalTrials.gov).

SUPPRESS-NSCLC is a phase II trial that randomly assigns patients with NSCLC who have evidence of oligo-progressive disease while on immune-checkpoint inhibitor or tyrosine kinase inhibitor regimens to SBRT plus current systemic medication or standard of care (NCT04405401, ClinicalTrials.gov).

### 4.3. Oligo-Persistent NSCLC

A phase II non-randomized prospective study (ATOM) tried to assess the efficacy of SBRT to oligo persistent lesion after 3 months of EGFR TKI treatment. Eighteen patients with ≤4 lesions were enrolled from 2014 to 2017; recruitment was stopped before the planned number of 34 because of slow accrual. The 1-year PFS was 68.8%, and there was no grade 3 or more toxicity [45].

Unfortunately, to the best of our knowledge, no prospective trials are being conducted in the sub-population of patients with oligo-recurrent NSCLC. This population still needs to be investigated in future clinical trials.

## 5. Breast Cancer

Breast cancer (BC) is the most common cancer in women worldwide, as well as the second leading cause of cancer-related death. The majority of BC deaths are the result of a distant recurrence or metastatic disease. De novo oligometastatic BC accounts for approximately 6% of all cases of metastatic BC, and 20–30% of all early-stage BC will relapse on a distant site. Given the long natural history of some metastatic BC, particularly those with hormone receptor positive disease and bone-only metastases, it appears ideal to treat all oligometastases with local therapy [46,47].

Similarly, for patients with de novo oligometastatic disease (untreated primary tumor plus limited metastases) it is uncertain whether surgery to the primary with or without adjuvant local RT is better than systemic therapy alone.

There are also several treatment systemic options for oligometastatic BC, like chemotherapy, target therapy, immunotherapy, or a combination of these approaches which makes the oligoprogressive scenario an ideal opportunity for incorporating metastasis-directed SBRT [48,49,50,51,52,53,54].

A study of 361 “all comers” extracranial oligometastatic cancer patients treated with SBRT sought to identify prognostic pretreatment factors to identify which patients may benefit the most from MDT [55]. Median OS was 47.1 months, with BC patients having significantly longer OS than colorectal, gastrointestinal, NSCLC, sarcoma, and other primary tumor types (Table 3, Figure 2). 

### 5.1. De Novo Oligometastatic BC

Locoregional treatments for primary breast tumors led to incongruent results that did not clearly identify a population that would benefit from breast surgery [56,57,58].

From 2005 to 2013, 716 women with de novo metastatic BC were randomized to receive locoregional treatment with surgery and adjuvant radiation at Tata Memorial Hospital. Median OS was not different between the two groups (19.2 vs. 20.5 months, *p* = 0.79). However, this study’s systemic therapy was criticized (e.g., limited taxane use; 92% of patients with HER2 positive disease did not receive anti-HER2 therapy) [59].

Between 2007 and 2012, a Turkish study called MF07-01 randomized 274 treatment-naive patients with stage IV BC to receive locoregional treatment followed by systemic therapy or systemic therapy alone. The 3-year survival rates were similar in both groups (60 vs. 51%, *p* = 0.1). The locoregional group had a median survival of 46 months compared to 37 months in the systemic therapy group (HR = 0.66, 95% CI: 0.49–0.88, *p* = 0.005). Patients with positive estrogen and progesterone receptors (ER+/PR+), HER2 negative, younger than 55 years of age, and solitary bone-only metastases benefited from local therapy [60].

**Table 3 cancers-14-01152-t003:** Main results of SBRT in oligometastatic BC.

Author/Year	N^o^ of Patients/Primary/Oligometastatic State	Phase/Design/N^o^ of Lesions	Arms (Investigational vs. Control)	Primary Endpoint	Median PFS (Months)	Median OS (Months)	Toxicity (≥G3)
Miyata et al./2017 [61]	21 Oligo-recurrence	Retrospective ≤2 lesions	EBRT/SBRT	PFS	24	41	5%
Trovo et al./2018 [62]	54 Mixed OM	Phase II ≤5 lesions	SBRT-MDT	PFS	24	NR	0%
Milano et al./2018 [63]	48 Mixed OMD	Phase II ≤5 lesions	SBRT-MDT	FFDM	36	60	NA
David et al./2020 [64]	15 Mixed OMD	Prospective ≤3 lesions	SBRT-MDT	Feasibility	NR	NA	0%

N: number; OMD: oligometastatic disease; FFDM: freedom from widespread distant metastasis; EBRT: external body radiotherapy; SBRT stereotactic radiotherapy; MDT: metastasis directed treatment; NA: not available; NR: not reached.

The place of locoregional treatment in stage IV disease should thus be further investigated.

Several studies are underway in order to investigate the role of metastasis-directed SBRT in the de novo oligometastatic BC.

The NRG oncology group designed a phase II/III trial that will study the impact of metastases guided RT in patients with de novo oligometastatic BC (NRG-BR002, ClinicalTrials.gov). Three hundred sixty patients will be randomized between the continuation of their current planned systemic therapy at the discretion of the treating physician (Arm 1) or the Arm 1 treatment with the addition of SBRT to metastatic sites. The primary endpoints will be PFS and OS, the study was suspended in September 2021 for interim analysis (NCT02364557, ClinicalTrials.gov).

STEREO-SEIN is a large prospective trial (*n* = 280) aiming at PFS improvement, the experimental arm will receive SBRT to all metastases and the beginning of systemic treatment will be administered 2 to7 days after SBRT completion, while the active comparator will not get SBRT (NCT02089100, ClinicalTrials.gov).

CLEAR is a multicenter, single-arm, phase 2 trial that will investigate the role of local treatment in addition to endocrine therapy in ER-positive/HER2-negative oligo-metastatic de novo BC. One hundred and ten patients are expected to be enrolled until mid-2025 (NCT 03750396, ClinicalTrials.gov). 

Another phase II/III trial will test whether treating BC metastases with surgery or high-dose radiation improves survival (OS and PFS). Until the end of 2022, 360 participants will be randomized to MDT (SBRT or surgery) or continuation of systemic therapy (NCT02364557, ClinicalTrials.gov).

### 5.2. Oligo-Progressive, Oligo-Recurrent and Oligo-Persistent Breast Cancer

Similarly, to the de novo setting, evidence for other subtypes of oligometastasis is lacking in BC, as prospective trials are uncommon and the patient population is not well defined.

Miyata et al. investigated the place of RT in a group of 21 patients treated for an oligo-recurrent BC relapse. The second oligometastatic relapse occurred after a median of 24 months, and the OS was 41 months. Toxicities were mild with only one grade 3 acute toxicity. The authors came to the conclusion that RT directed to oligo-metastasis could differ in time to a new distant recurrence [61].

A prospective phase II multicentric trial was designed to determine if administering MDT to all metastatic sites could improve the PFS in patients with oligometastatic BC. Patients presented with BC with up to five metastatic sites, no brain metastases, and they presented a non-treated primary tumor. SBRT technique or fractionated intensity-modulated RT (IMRT) were permitted. Fifty-four patients with 92 metastatic lesions were included in the study. The one-year and two-year PFS rates were 75% and 53%, respectively. The two-year LC and OS rates were 97% and 95%, respectively. RT was well tolerated with no evidence of grade 3 toxicity. The authors came to the conclusion that radical radiation therapy to all metastatic sites should be used in patients with oligometastatic BC [62].

Milano et al. published an update on the results of a phase II nonrandomized prospective trial involving forty-eight women with 1–5 extracranial BC oligometastases who received SBRT to all radiographically visible sites of disease. After SBRT, the 5- and 10-year OS rates for patients who suffered from bone only metastasis were 83% and 75%, respectively, while for patients with visceral disease the 5- and 10-years OS rates were 31% and 17%, (*p* = 0.002). The tumor burden, the number of oligometastatic lesions and the presence of visceral metastasis were significant factors of freedom from widespread metastasis. This emphasizes that patients with BC with oligometastatic disease treated with SBRT can have a positive outcome but that this depends on the volume and the number of lesions as well as their location (visceral vs. bone) [63].

In the same line, David et al. reported the results of a single institution prospective trial on single fraction SBRT for patients with bone only oligometastatic BC. Each patient received 20 Gy in one fraction to each metastasis (1–3 lesions) [64]. The two-year LC was 100% and the PFS was 67%. SBRT was safe and effective in this cohort, with two-thirds of the patients disease-free after two years. No patients experienced a grade 3 or more toxicity [64].

The phase II randomized CURB trial has evaluated the benefit of SBRT in metastatic NSCLC and BC. However, the 12-week PFS benefit was only found in NSCLC patients, and not in BC patients. It is necessary to continue to investigate the role of metastasis guided SBRT in BC and why there may be a difference in benefit between cohorts depending on the primary [65].

## 6. Renal Cell Carcinoma

About 16% of renal cell carcinoma (RCC) patients present with locally advanced or de novo metastatic disease at diagnosis for which surgery is not feasible [66]. The natural history of advanced or metastatic RCC varies from months to years depending on clinical, pathologic, laboratory, and radiographic features [67].

Depending on the extent of disease, sites of involvement, and patient-specific factors, systemic therapy (immunotherapy, molecularly targeted agents), surgery, and RT may all play a role.

Systemic therapy is the cornerstone of treatment for de novo metastatic RCC, and new guidelines adapted from the European Association of Urology (EAU) and European Society of Medical Oncology (ESMO) and based on the International Metastatic RCC Database Consortium (IMDC) risk classification agreed that different combinations of immunotherapy and anti-angiogenic therapy must be offered upfront to newly diagnosed metastatic RCC [68] (Table 1).

### 6.1. De Novo Oligometastatic RCC

Over the past decade, evidence suggests that in oligometastatic RCC aggressive local therapy could improve outcomes. RCC was historically known to be radio-resistant to conventional RT; however, important clinical responses have been observed in patients treated with SBRT which serves to reinforce the concept that RCC may not be as radioresistant as previously thought [69]. For instance, a meta-analysis of 28 studies looked at the role of SBRT in the treatment of oligometastatic RCC [70]. There were 679 patients with a total of 1159 extracranial lesions. The median treatment volume was 59.7 cc, the 1-year LC rate was 89.1% and the 1-year survival rate was 86.8%. For extracranial disease, the incidence of any grade 3–4 toxicity was 0.7%.

Moreover, in the metastatic setting in patients receiving ≤2 prior anti-angiogenic therapies, the non-randomized phase II NIVES study tested the combination of SBRT (3 fractions of 10 Gy each) concomitant to immunotherapy (nivolumab, anti-PD1, 240 mg every 14 days for 6 months). Sixty-nine patients were enrolled. The overall response rate (ORR) was 17% and the disease control rate was 55%. The median PFS was 5.6 months (95% CI, 2.9–7.1) and median OS 20 months (95% CI, 17-not reached). After 1.5 years of follow-up, 23 patients died. The median duration of response was 14 months. No new safety concerns arose. [71]. In the same context, the RADVAX trial investigated if patients with metastatic RCC receiving nivolumab and ipilimumab (anti-CTLA-4 antibody) benefited from the addition of SBRT (five fractions of 10 Gy each) to 1–2 metastatic sites administered between the first and second dose of immunotherapy. The ORR in the 25 enrolled patients was 56%, while two grade 2 toxicities were observed [72].

In the UT Southwestern phase II single arm study, 47 patients with de novo oligometastatic RCC, were treated with SBRT on 88 extracranial lesions prior to starting systemic therapy [73,74]. The LC rate was 91.5% at two years, with no grade 3 toxicity. The median time to start systemic therapy was 15.2 months and the percentage of patients with no metachronous illness at 1 year improved significantly. The same group of investigators is now planning a prospective trial that will investigate the role of SBRT in this particular population of de novo oligometastatic RCC. The primary endpoint is the time to start systemic therapy, 23 patients are expected to be enrolled until the end of 2023 (NCT02956798, ClinicalTrials.gov).

These studies show that SBRT is safe and effective in RCC, and its use should be tailored to specific situations, such as when a patient is oligometastatic and the treating physician wants to postpone a change in systemic therapy, or when a patient cannot receive either anti-angiogenic therapy, immunotherapy, or a combination of the two.

In some cases, patients progressing on immunotherapy required SBRT due to oligoprogression; in this case, SBRT should not be used to look for an abscopal effect, which is a rare event in itself, but rather to reduce metastatic tumor burden.

Similarly, SBRT should be considered as an alternative to surgery or invasive MDT in patients with metastatic RCC and untreated primary tumors.

### 6.2. Oligo-Progressive RCC

Santini et al. retrospectively enrolled 55 patients who experienced disease oligo-progression after at least 6 months from the beginning of first-line therapy and were treated with MDT. The median time to the first relapse after MDT was 14 months. Patients who received the same therapy after SBRT treatment on a site of progression had significantly longer OS (from the time of first oligo-progression) than those who switched therapies (39 vs. 11 months, *p* = 0.014) [75].

Prospective trials are being developed to assess the role of SBRT in oligo-progressive RCC patients. GETUG-StORM is a multicenter phase II prospective trial that will investigate the efficacy of SBRT in prolonging PFS in patients with oligo-metastatic RCC and in which proportion it can delay the initiation of systemic therapy. Patients will be randomized 2:1, 114 patients are expected to be included until the end of 2023 (NCT02956798, ClinicalTrials.gov). Another single institution trial will enroll patients in the same setting with a similar endpoint (patients under first line Sunitinib), 38 participants will be enrolled (NCT02019576, ClinicalTrials.gov).

### 6.3. Oligo-Persistent RCC

A single arm phase II study looked at the feasibility of SBRT to all metastatic sites that remained after a first line of systemic treatment for RCC (TKI or immunotherapy) and in patients with de novo oligometastatic disease. Patients could be treated on up to five lesions, and they had to stop their systemic treatment at least one month before SBRT. Thirty patients were enrolled, the median PFS was 22.7 months and no grade 3 complications were observed. The authors concluded that sequential RT could defer systemic therapy and allow for systemic therapy breaks in patients with oligo-persistent RCC [76] (Table 4).

## 7. Colorectal Cancer (CRC)

Colorectal cancer is the third most common cancer affecting both males and females in Europe. Approximately 20 to 25% of newly diagnosed colon cancers are metastatic at presentation (synchronous metastasis). Others may develop metastatic disease after potentially curative treatment of localized disease. The most common distant metastatic sites are the liver, lungs, lymph nodes, and peritoneum.

Despite significant advances in systemic chemotherapy that have increased median survival from less than one year in the single-agent fluoropyrimidine era to more than 30 months, fewer than 20% of those treated with chemotherapy alone are still alive at five years, and only a few are disease-free unless metastasis resection or ablation is timely offered [77]. Surgery, on the other hand, offers a potentially curative option for selected patients with limited metastatic disease, most commonly in the liver and lung. Metastasectomy can result in long-term survival in up to 50% of cases, and an aggressive surgical approach to both the primary and metastatic sites is required in conjunction with systemic chemotherapy. However, even after complete resection of metastases, the majority of patients who survive five years have the active disease; only about 20 to 30% are free of recurrence long-term and may be cured.

To classify patients with oligometastatic CRC with liver metastasis, Pitroda et al. developed an integrated molecular classification based on the analysis of 134 patients that benefited from liver metastasectomy. Three subtypes were defined (low risk, intermediate risk, and high risk patients), with 10-year OS rates of 94%, 45%, and 19%, respectively [78]. Poor-prognosis subtypes have VEGFA amplification in conjunction with stromal, mesenchymal, and angiogenic signatures (Subtype 3 stromal), or exclusive *NOTCH1* and *PIK3C2B* mutations with E2F/MYC activation (Subtype 1 canonical) [78]. 

To the best of our knowledge, no prospective or retrospective studies have demonstrated the outcome of patients based on the subset of oligometastatic disease they have.

More pertinent data can be found in the meta-analysis presented here.

Patients with oligometastasis in the liver are typically offered surgery. SBRT is usually offered to treat inoperable lesions. A meta-analysis of 18 studies involving 656 patients with oligometastatic CRC and treated by SBRT on the liver found a one-year OS of 67.2% (95% CI: 42.1–92.2) and a two-year OS of 56.5% (95% CI: 36.7–76.2), respectively. The median PFS and OS were 11.5 and 31.5 months, respectively. The pooled one-year LC was 67.0% (95% CI: 43.8–90.2), and the pooled two-year LC was 59.3% (95% CI: 37.2–81.5). Mild-moderate and severe liver toxicity were 30.7% and 8.7%, respectively. SBRT for liver oligometastases is an effective treatment option for patients with advanced CRC, with encouraging LC and survival [79] (Table 1).

The largest multicenter retrospective study on the topic of lung oligometastasis from CRC primary showed a 75.4% LC rate at 2 years after analyzing the outcomes of 1033 lesions. LC was significantly improved when the lesion was less than 2 cm in diameter and the treatment dose was at least 125 Gy BED (biological equivalent dose). A meta-analysis attempted to summarize the outcomes of SBRT for CRC lung oligometastases. Eighteen retrospective studies with a total of 1920 patients found that the LC rate in patients with CRC pulmonary oligometastases was significantly lower than in patients with other cancers (odds ratio 3.10, *p* = 0.00001) (Table 1, Figure 2) [80].

## 8. Future Developments of RT in Oligometastatic Disease in Mixed Primaries

After publishing encouraging results from the SABR-COMET phase II trial, the authors concluded the necessity to confirm their results through a larger phase III study [11]. COMET-10 is a prospective phase III trial that will include patients presenting 4–10 metastatic lesions [81]. One hundred and fifty-nine patients will be randomized to receive standard-of-care palliative-intent treatments (control arm), or standard-of-care treatment and SBRT to all sites of known disease (SBRT arm). This study will provide an assessment of the impact of SBRT on clinical outcomes and quality-of-life, to determine if long-term survival can be achieved for selected patients with 4–10 oligometastatic lesions. The trial is registered on ClinicalTrials.gov, NCT03721341, and started on 22 February 2019, the completion date is estimated for January 2029.

SABR-5 is a population-based phase II trial of SBRT for up to five oligometastases. In this non-randomized phase II trial, all participants will receive experimental SBRT treatment to all sites of newly diagnosed or progressing oligometastatic disease. Two hundred patients will be enrolled (expected primary completion 2025) to assess toxicity associated with this experimental treatment and measure late grade 4 toxicity [82]. 

EORTC is also conducting a phase III superiority study (OligoRARE, ClinicalTrials.gov) comparing the effect of adding SBRT to the standard of care treatment on OS in patients with rare oligometastatic cancers. Patients will be randomized in a 1:1 ratio between current standard of care treatment vs. standard of care treatment + SBRT to all sites of known metastatic disease. The primary objective of this trial is to assess if the addition of SBRT to the standard of care treatment improves OS as compared to standard of care treatment alone in patients with rare oligometastatic cancers. Two hundred patients will be recruited from June 2021 to August 2028. Expected results in 2030 should assess the results of SABR-COMET cohorts.

## 9. Future Directions

The term oligometastatic disease originated in the 1990s, during a period of less precise radiation technology. With the possibility of a patient having multiple metastases that can be treated concurrently or sequentially with SBRT, the concept of oligometastases should be reconsidered.

For example, in neuro-oncology, stereotactic radiosurgery is now possible to treat more than 10 to 15 individual lesions concurrently without experiencing significant toxicities [83,84].

What we do not know is the maximum number of body metastases that can be treated safely in this manner. Numerous constraints currently prevent the widespread use of multisite SBRT. When high-dose RT is delivered to multiple targets, the first constraint is a lack of knowledge regarding appropriate cumulative dose constraints. While current dose-volume histogram measurements and constraints provide excellent detail on doses to critical organs, scatter dose (e.g., volumes receiving 5–10 Gy) may become significant issues in plans with multiple targets, such as at the level of hematological toxicity.

Multi-site irradiation should be evaluated from a feasibility standpoint because it requires repeated measurements, contouring, as well as longer treatment times. Numerous tools and techniques are being developed to assist in the development of a suitable workflow. New technologies in development (e.g., Ethos^TM^, Varian Medical System, for adaptative radiotherapy using artificial intelligence, or RayIntelligence^®^, RaySearch Laboratories AB, using deep learning analytics system) may enable more rapid treatment of multiple sites, thereby, reducing planning and treatment time and mitigating the effects of target motion and uncertainty [85,86]. Auto-contouring, auto-planning, and auto-quality assurance tools, for example, can significantly reduce the time to prepare and deliver treatments [87].

As a result, we believe that the unanswered research question in our field is now whether SBRT is safe and feasible when administered concurrently or sequentially to multiple (>5) metastatic sites. If proven to be safe, multi-site SBRT would be an ideal complement to the growing effectiveness of systemic treatments.

## 10. Conclusions

Stereotactic body radiation therapy is a type of radiation therapy that involves delivering a small number of high doses of radiation to a target volume; this technique has quickly gained popularity due to its excellent tolerability and high loco-regional control rates that approach 90%. The routine use of SBRT necessitates careful consideration of organ motion, and adaptive technology is constantly evolving to make these treatments more precise, resulting in fewer and fewer side effects.

To date, evidence for treating oligometastasis is based on the proof-of-concept results of the SABR- COMET randomized trial, which demonstrated a significant increase in OS with the use of SBRT compared to best supportive care.

We believe that future studies should focus on randomizing patients based on disease type and stratifying patients based on the number of metastases, as well as disease location (bone vs. visceral), as these are distinct prognostic factors that should be considered.

New guidelines have established the use of specific nomenclature: oligorecurrence, oligoprogression, and oligopersistence taking into account whether oligometastatic disease is diagnosed during a treatment-free interval or during active systemic therapy, as well as whether an oligometastatic lesion is progressing on current imaging. This oligometastatic disease classification and nomenclature should be evaluated in clinical trials in the future because each disease subtype and oligometastatic state may have different outcomes.

SBRT could then be used in a dynamic conception of oligometastatic disease, and if proven to be safe, multi-site SBRT would be an ideal complement to the growing effectiveness of systemic treatments. Efforts should also be made to obtain paired pre-post treatment tumor biopsies in order to determine which patients benefit the most from SBRT.

The concept of oligometastatic disease, as well as the implementation of SBRT, should not be viewed in isolation from the current systemic treatment that a given disease type requires, and its incorporation into standard cancer patient management will require not only prospective randomized trials but synergistic multidisciplinary teams capable of evaluating patients on a case-by-case basis and deciding when and how to incorporate SBRT in a given clinical scenario.

## Figures and Tables

**Figure 1 cancers-14-01152-f001:**
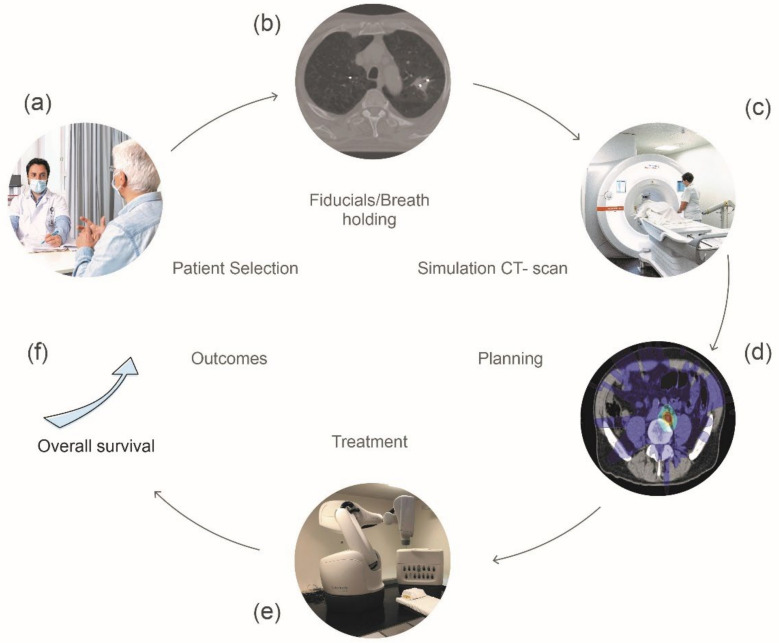
Steps for planning a stereotactic body radiotherapy (SBRT) treatment. (**a**) Consultation with the patient to address the treatment goals and potential side effects, (**b**) Placement of fiducial markers for tumor tracking purposes all along the treatment, (**c**) 3D imaging CT scan that provides tumor’s precise location, (**d**) Treatment dosimetry planning provides dosage level and positioning of radiation beams, (**e**) Treatment is delivered with sophisticated machines that allow for daily imaging to ensure proper tumor positioning, (**f**) Potential increase in OS observed in some studies.

**Figure 2 cancers-14-01152-f002:**
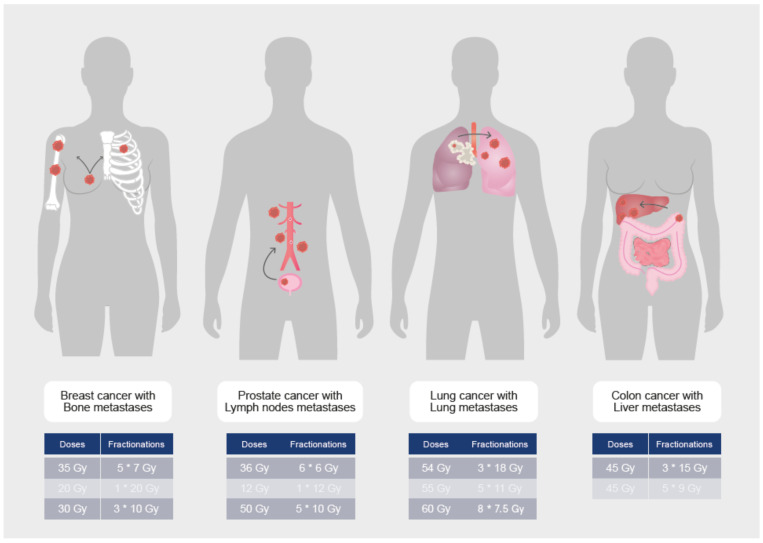
SBRT dose and fractionation depending on metastatic sites.

**Figure 3 cancers-14-01152-f003:**
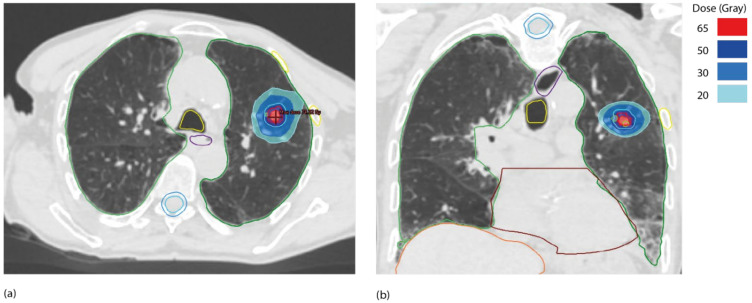
Dosimetry of a 55 Gy in five fractions of 11 Gy lung SBRT for pulmonary metastasis from lung cancer primary. (**a**) axial view, (**b**) coronal view. Gross tumor volume (GTV) outlined in red, planning target volume (PTV) outlined in blue and fiducial markers outlined in green.

**Table 4 cancers-14-01152-t004:** Main results of SBRT in oligometastatic RCC.

Author/Year	N^o^ of Patients/Primary/Oligometastatic State	Phase/Design/N^o^ of Lesions	Arms (Investigational vs. Control)	Primary Endpoint	Median PFS (Months)	Median OS (Months)	Toxicity (≥G3)
Zhang et al./2019 [74]	47 De novo OMD	Retrospective ≤4 lesions	SBRT-MDT	FST	15	NR	0%
Santini et al./2017 [75]	55 Oligo-progressive	Retrospective ≤5 lesions	MTD	PFS	14	37	NA
Tang et al./2021 [76]	30 De novo/Oligo-progressive	Phase II ≤5 lesions	SBRT-MDT	Feasibility	23	NR	10%

N: number; OMD: oligometastatic disease; FST freedom from systemic therapy; EBRT: external body radiotherapy; SBRT stereotactic radiotherapy; MDT: metastasis directed treatment; NA: not available; NR: not reached.

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
