# Peer review of "Stereotactic Body Radiation Therapy in Patients with Oligometastatic Disease: Clinical State of the Art and Perspectives"

_cancers, 2022, doi:10.3390/cancers14051152_

Round 1

Reviewer 1 Report

First of all, I would like to congratulate you on this work, which deals with a subject of increasing interest to all of us.

I appreciated the schematic organization of the paper, and certainly also the use of the new nomenclature.

I would like to give some advice that could make easier the reading.

- I believe that the part concerning systemic therapy in the introductory section of prostate cancer (line 128-150) is a little too detailed for the purpose of the work, particularly compared to the others, and could make the reading difficult.

- I would put the reference to Tab.1 of line 160 in the general part

- Line 508: use acronym MDT instead of its extended form

- Finally, to streamline the discussion, you could be consider the hypothesis of inserting the part of the ongoing trials in a separate section, perhaps linked to the paragraph on future perspectives

Reviewer 2 Report

Dr Kinj et al. have submitted a systematic review to summarize the available evidence and future perspectives on the role of SBRT in 5 types of oligometastatic cancers. This is a well-conducted study and data are presented logically and clearly. The level of evidence is good to support SBRT as an important tool for the treatment of specific oligometastatic cancers, in different settings and in a multidisciplinary approach. I think this review will be of interest to the readership. 

I just have few minor issues:

  • Table 1 shows data and articles about all types of cancers reported by authors; it could be useful to split it in more tables, one for each type of tumor, and to insert each one in the appropriate paragraph;
  • Data about toxicity are presented across the text: Table 1 could be improved reporting data about toxicity, where applicable, in order to favour a better reading and quick comprehension of data.
  • This review deals with SBRT: title, reporting generically "Radiotherapy", could be misleading and should be changed in "Stereotactic Body Radiation Therapy (SBRT) in patients with oligometastatic disease: clinical state of the art and perspectives".

Author Response

Reviewer # 2 - general comments:

Dr Kinj et al. have submitted a systematic review to summarize the available evidence and future perspectives on the role of SBRT in 5 types of oligometastatic cancers. This is a well-conducted study and data are presented logically and clearly. The level of evidence is good to support SBRT as an important tool for the treatment of specific oligometastatic cancers, in different settings and in a multidisciplinary approach. I think this review will be of interest to the readership.

 I just have few minor issues:

Reply to reviewer # 2 - general comments:

We thank the reviewer for the positive feedback, and for his excellent comments and advice which has tremendously helped us to improve the clarity and content of the manuscript.

Reviewer # 2 - comment 1:

Table 1 shows data and articles about all types of cancers reported by authors; it could be useful to split it in more tables, one for each type of tumor, and to insert each one in the appropriate paragraph;

Reply to reviewer # 2 comment 1:

We thank the reviewer for this idea. We have spitted the Tables as suggested by the reviewer. There are now 4 Tables inserted at the end of each disease type and summarizing the clinical studies.

Reviewer # 2 comment 2:

Data about toxicity are presented across the text: Table 1 could be improved reporting data about toxicity, where applicable, in order to favour a better reading and quick comprehension of data.

Reply to reviewer # 2 - comment 2:

We thank the reviewer for this observation. We have added the toxicity to each of the Tables as suggested by the reviewer.

Reviewer #2 - comment 3:

This review deals with SBRT: title, reporting generically "Radiotherapy", could be misleading and should be changed in "Stereotactic Body Radiation Therapy (SBRT) in patients with oligometastatic disease: clinical state of the art and perspectives".

Reply to reviewer # 2 - comment 3

We thank the reviewer for noticing this. We agree that SBRT should be mentioned in the Title. We have thus accepted this suggestion and the new title is “Stereotactic body radiation therapy  in patients with oligometastatic disease: clinical state of the art and perspectives”.
